Influence of substrate types and morphological traits on movement behavior in a toad and newt species

http://orcid.org/0000-0003-3447-977X Trochet Audrey 1 2 audrey.trochet@sete.cnrs.fr
Le Chevalier Hugo 1 2
Calvez Olivier 2
Ribéron Alexandre 1
Bertrand Romain 2 3
Blanchet Simon 1 2
1 Laboratoire Evolution et Diversité Biologique, Université Paul Sabatier (Toulouse III) , Toulouse , France
2 Station d’Ecologie Théorioque et Expérimentale, CNRS , Moulis , France
3 Center for Biodiversity Theory and Modelling, CNRS , Moulis , France
Kramer Donald
Electronic publication date: 2019 Jan 9
Publication date: 2019
Volume: 6
Electronic Location ID: e6053
Received 2018 Jul 2; Accepted 2018 Oct 29
Copyright: © 2019 Trochet et al.
Copyright year: 2019
Copyright holder: Trochet et al.
License: This is an open access article distributed under the terms of the Creative Commons Attribution License, which permits unrestricted use, distribution, reproduction and adaptation in any medium and for any purpose provided that it is properly attributed. For attribution, the original author(s), title, publication source (PeerJ) and either DOI or URL of the article must be cited.
License URL: https://creativecommons.org/licenses/by/4.0/

Keywords: Matrix permeability, Inter-patch movements, Fragmented landscapes, Roads, Common toads, Bufonidae, Marbled newts, Connectivity, Salamandridae

Funding: Fondation de France French Laboratory of Excellence project “TULIP” ANR-10-LABX-41; ANR-11-IDEX-0002-02 “Investissement d’Avenir” grants managed by Agence Nationale de la Recherche AnaEE, ANR-11-INBS-0001AnaEE-Services Financial support was provided by a post-doctoral fellowship to Audrey Trochet from the Fondation de France. This work was supported by the French Laboratory of Excellence project “TULIP” (ANR-10-LABX-41; ANR-11-IDEX-0002-02) and benefited from both “Investissement d’Avenir” grants managed by Agence Nationale de la Recherche (AnaEE, ANR-11-INBS-0001AnaEE-Services). The funders had no role in study design, data collection and analysis, decision to publish, or preparation of the manuscript.

==============================
Background

Inter-patch movements may lead to genetic mixing, decreasing both inbreeding and population extinction risks, and is hence a crucial aspect of amphibian meta-population dynamics. Traveling through heterogeneous landscapes might be particularly risky for amphibians. Understanding how these species perceive their environment and how they move in heterogeneous habitats is an essential step in explaining metapopulation dynamics and can be important for predicting species’ responses to climate change and for conservation policy and management.

Methods

Using an experimental approach, the present study focused on the movement behavior (crossing speed and number of stops) on different substrates mimicking landscape components (human-made and natural substrates) in two amphibian species contrasting in locomotion mode: the common toad (Bufo bufo), a hopping and burrowing anuran and the marbled newt (Triturus marmoratus), a walking salamander. We tested the hypothesis that species reaction to substrate is dependent on specific ecological requirements or locomotion modes because of morphological and behavioral differences.

Results

In both species, substrate type influenced individual crossing speed, with individuals moving faster on soil than on concrete substrate. We also demonstrated that long-legged individuals moved faster than individuals with short legs. In both species, the number of stops was higher in females than in males. In common toads, the number of stops did not vary between substrates tested, whereas in marbled newts the number of stops was higher on concrete than on soil substrate.

Discussion

We highlighted that concrete substrate (mimicking roads) negatively affect the crossing speed of both studied species, with an effect potentially higher in marbled newts. Our findings corroborate negative effects of such heterogeneous landscapes on movement behavior of two amphibian species, which may have implications for the dynamics of metapopulations.

Introduction

Inter-patch movements are key processes for maintaining gene flow among populations (Kareiva & Wennergren, 1995; Ronce, 2007) with strong consequences for metapopulation dynamics and population persistence (Clobert et al., 2001, 2012; Bowler & Benton, 2005). For decades, the architecture of landscapes has profoundly changed due to human activities. Anthropogenic practices, such as agriculture, urbanization, or the expansion of road networks, have led to discontinuities in the habitat matrix: formerly continuous habitats become smaller and more isolated from each other, resulting in the well-known habitat fragmentation pattern (Collinge, 2009; Wilson et al., 2016). Fragmentation and land use change could create inhospitable habitat within the current mobility range of individuals, which may force them to adapt their movement behavior (Arendt, 1988; Andreassen & Ims, 1998; Kuefler et al., 2010) to ensure sufficient connectivity among populations in spite of these environmental changes. This change in movement behavior may increase the costs associated with mobility, for instance by exposing individuals to higher mortality rate during the transience phase (when crossing roads, e.g.; Carr, Pope & Fahrig, 2002). Elucidating how individuals react and adapt their movement patterns in disturbed landscapes might improve our knowledge of metapopulation dynamics across a changing landscape, increase the realism of predictions of species’ responses to global change, and support the implementation of pertinent conservation plans.

One-third of the amphibian species are currently threatened worldwide, with 43% of species having declined over the last few decades (Stuart et al., 2004). Due to ecological requirements, pond- and stream-breeding amphibians are exposed to different habitat types (i.e., both terrestrial and aquatic) throughout their life cycle, often in patchy and heterogeneous landscapes (Marsh & Trenham, 2001). During the terrestrial phase, individual movements are risky, due to through predator and UV-B exposure (Kats et al., 2000) and desiccation risk (Rothermel & Luhring, 2005; Pittman et al., 2013; Pittman, Osbourn & Semlitsch, 2014). Many studies have considered the multiple effects of habitat fragmentation—and their related landscape components—on amphibian populations, both at the individual and population levels (see Cushman, 2006 for a review). In particular, urban areas and human-made infrastructure, such as roads, negatively affect these species in two ways: directly, through fatal collisions with vehicles (Fahrig et al., 1995; Rytwinski & Fahrig, 2015) and indirectly by fragmenting habitat and subsequent reduction of gene flow and colonization events (DeMaynadier & Hunter, 2000; Mazerolle, 2004; Marsh et al., 2005; Cosentino, Schooley & Phillips, 2011; Youngquist et al., 2016; Lenhardt et al., 2017). Amphibians are more susceptible to fatal collisions when crossing roads during migration (Glista, DeVault & DeWoody, 2008) because they often become immobile when facing an approaching vehicle (Mazerolle, Huot & Gravel, 2005; Bouchard et al., 2009). Also, the mobility of crossing individuals could be affected by the nature of the road’s substrate. In this vein, several studies investigated the effects of different substrates on individual movements at the intra-specific level (Ims & Yoccoz, 1997; Wiens, Schooley & Weeks, 1997; Wiens, 2001; Stevens et al., 2006; Semlitsch et al., 2012). For instance in Plethodon metcalfi, the individual crossing speed was higher on asphalt than on grass (Semlitsch et al., 2012) with various consequences on mobility success. Some amphibian species can also benefit from anthropogenic landscape elements. For example, Rhinella marina (cane toads) seemed to use roads as dispersal corridors in Australia (Brown et al., 2006). Despite the crucial importance of inter-patch movements in altered landscapes, little is known about the multi-species interactions among individual movements and different substrates (but see Rothermel & Semlitsch, 2002). Indeed, habitat-species interactions are complex and highly specific (Kolozsvary & Swihart, 1999; Trochet et al., 2016a). Heterogeneous landscapes and associated landscape parameters could differently alter movement of organisms with strong divergence between species (Rothermel & Semlitsch, 2002) resulting in contrasting conservation needs and species suitability under global change.

The costs associated with inter-patch movement can be high (Van Dyck & Baguette, 2005) and could lead to high selective pressures on mobility and associated phenotypic traits (Bonte et al., 2012). According to this expectation, many studies have focused on the correlation between movement and phenotypic traits. At the intra-specific level, phenotypic differences related to dispersal ability between individuals have been reported. For instance, larger and/or longer individuals are generally expected to be dispersers, because they should benefit from their high levels of competition to disperse farther (Léna et al., 1998). Evidence for a relationship between body size and movement has been described in many taxa (insects: Anholt, 1990; Legrand et al., 2015; mammals: Gundersen, Andreassen & Ims, 2002; Holekamp & Sherman, 1989; O’Riain, Jarvis & Faulkes, 1996; reptiles: Léna et al., 1998; birds: Barbraud, Johnson & Bertault, 2003; Delgado et al., 2010; fishes: Radinger & Wolter, 2014; amphibians: Ousterhout & Semlitsch, 2018). For walking and/or hopping animals, selection for efficient displacement might lead to leg elongation. As a result, morphological adaptation to movement can be deduced from estimates of leg length (Moya-Laraño et al., 2008). This correlation between movement and leg length (i.e., hind-limb length, hereafter HLL) has been demonstrated in some species (reptiles: Losos, 1990; spiders: Moya-Laraño et al., 2008; amphibians: Bennett, Garland & Else, 1989; Choi, Shim & Ricklefs, 2003; Phillips et al., 2006), but still remains rare.

Understanding how amphibians perceive their environment and how they actually move in heterogeneous habitats is an essential step for understanding metapopulation structure, and can be important for improving the realism of predictive models of species’ responses to global change. The present study focused on the movement behavior (crossing speed) on two different substrates mimicking landscape components (a human-made substrate and a natural substrate) in two contrasting amphibian species, the common toad (Bufo bufo), a hopping and burrowing anuran, and the marbled newt (Triturus marmoratus), a walking salamander. Each species has specific ecological requirements: common toads have an explosive breeding season and spend a large proportion of their life on terrestrial habitats (i.e., forests, bushlands, or urban areas) whereas the breeding season of marbled newts is longer, and this species inhabits relatively small well-vegetated ponds surrounded by woodlands (Jehle & Arntzen, 2000; AmphibiaWeb, 2012). Contrary to common toads that generally avoid this kind of habitat type, marbled newts also occur in more open areas like heathens and agricultural landscapes. Common toads and marbled newts can live within the same habitat, such as grassland or woodland, and could therefore face the same environmental pressures during terrestrial movements (Daversa, Muths & Bosch, 2012; Trochet et al., 2017). Also, both the contrasting morphology and modes of locomotion induce different muscular contractions during displacement, leading to different mobility ability between the two species (Smith & Green, 2005) and therefore explain the selection of various habitats encounter during terrestrial phase.

Our study aimed at testing the influence of different substrate types on two amphibian species in order to highlight if and how substrates can alter movement in these sympatric species. Considering these species allowed testing the hypothesis that species could react differently to substrate nature, depending on specific ecological requirements, mobility abilities, or locomotion modes because of morphological and behavioral differences. Identifying such differences could help improving our understanding about species-specific interactions within a human-made environment. Consequently, it would be interesting to consider those interactions to improve predictive models of species’ responses to climate change and to propose efficient conservation management plans.

Materials and Methods

Studied species

The common toad is one of the most widely distributed and abundant anuran species in Europe (Gasc et al., 1997). Usually, reproduction occurs in February, and large numbers of toads migrate to breeding sites (i.e., large ponds, ditches, or lakes with relatively clear water, quite variable in area and depth; AmphibiaWeb, 2012) where the males compete for mating. After an explosive breeding season, toads leave ponds and return to terrestrial habitats (Gittins, 1983) where they spend a large proportion of their life. This species occupies not only various terrestrial habitats over three to four km from the breeding site (Smith & Green, 2005), such as coniferous, mixed and deciduous forests, bushlands, but also urban areas such as gardens and parks (Nollërt & Nollërt, 2003). Common toads hibernate singly or in groups from September to February, on land and occasionally in streams and springs.

The marbled newt is a large-bodied urodele species from Western Europe, found in France, Spain, and Portugal (Sillero et al., 2014). The reproduction period is longer than common toads (from the beginning of March to July) and takes place in different aquatic habitats, including well-vegetated ponds, pools, ditches, and streams (AmphibiaWeb, 2017). After breeding, adults leave water bodies by walking, and deciduous or mixed woodland, where they find refuges under dead and rotting wood and other hiding places (Jehle & Arntzen, 2000). Displacement distances of newts around their breeding sites are shorter than anurans and in the range of several hundred meters (Trochet et al., 2017).

Sampling and morphological measurements

Our work complies with the international animal care guidelines of the Association for the Study of Animal Behaviour, and all required French permits relating to authorization of capture, marking, transport, detention, use, and release of protected amphibian species have been obtained (permit nos. 09-2014-14 and 32-2014-07; animal experimentation accreditation no. A09-1) from the DREAL Occitanie (“Direction Régionale de l’Environnement, de l’Aménagement et du Logement”). Ethical approval was included under the protected species handling permit from the DREAL Occitanie. The project was approved by the “Conseil National de la Protection de la Nature” on September 14, 2014 and by the “Conseil Scientifique Régional du Patrimoine Naturel (CSRPN)” of the region Midi-Pyrénées on October 14, 2014.

In total, 83 common toads (68 males and 15 non-gravid females) and 46 marbled newts (23 males and 23 non-gravid females) were captured in the south of France at the end of breeding season (for toads: from March 13, 2015 to March 20, 2015; for newts: from April 28, 2015 to May 6, 2015) within and near two different ponds to reduce potential impact on populations (geographical coordinates of pond 1 43.671781°N, 0.504308°E and pond 2: 43.076347°N, 1.351639°E). Individuals were then brought to the lab for experimentation and released between June and July 2015. During experiments, animals were housed at the Station d’Ecologie Théorique et Expérimentale (Moulis, France) in same-species groups of four to six individuals in semi-aquatic terraria of 60 × 30 × 30 cm at 20 °C. They were fed ad libitum with live mealworms and tubifex worms. For unambiguous identification, all individuals were PIT-tagged (RFID Standards ISO 11784 & 11785 type FDX-B, 1.4 × 8 mm, 134.2 khz from BIOLOG-ID, France; animal experimentation accreditation no. A09-1) immediately after capture and before the experiments following the protocol developed in Le Chevalier et al. (2017). We then measured snout-to-vent length (SVL) and HLL to the nearest one mm and body weight (mass) to the nearest 0.01 g. To limit the effects of stress on behavioral responses, all individuals were kept in captivity for several days without any manipulation until the experimental tests.

Movement tests

All tests were performed in June and July 2015, after the breeding season when all individuals were in the terrestrial phase. In order to test the crossing capacities of both species, we made them move along two tracks (200 cm long × 10 cm wide × 20 cm high), each filled with a different substrate: smooth concrete slab (human-made) or soil (natural). Tracks were not moistened so that the substrate was unfavorable. During the experiments, all individuals were chased down the tracks and forced to move by gently poking their back with a finger after each stop (3 s between pokes if needed). Only one individual was tested at a time and we recorded the number of stops (stops) and the crossing speed (in cm/s; including stops) to the nearest 0.1 s to travel 200 cm from departure to arrival line. In order to provide reliable estimates of crossing capacity using a repeated-measure design while minimizing stress, every individual was tested three times on each substrate. In toads, tests were randomly spread over 2 days (three trials on day 1 and three trials on day 2; from March 18, 2015 to March 31, 2015) while newts, for which we kept animals during a longer period for another experiment not detailed here, tests were randomly performed over 27.9 ± 8.5 days (mean ± SD; from April 28, 2015 to June 2, 2015). Outside of test periods, animals were returned to the semi-aquatic aquaria. Because locomotion in amphibians is influenced by temperature (Herrel & Bonneaud, 2012; James et al., 2012; Šamajová & Gvoždík, 2010), all tests were performed in a greenhouse under controlled-temperature conditions (mean ± SD: 25 ± 1 °C) with light conditions similar to nature.

Some individuals (n = 32) did not complete the 200 cm run (stopping completely—three successive pokes without moving—or turning back); these individuals and their replicates were removed from the analyses for statistical reasons. We therefore included 77 toads (77 toads × 3 replicates × 2 substrates = 462 tests) and 20 marbled newts (20 newts × 3 replicates × 2 substrates = 120 tests) in the analyses.

Statistical analyses

For each species, the influence of substrate (concrete and soil) on crossing speed was first tested using Wilcoxon rank-sum tests and then modeled using linear mixed-effect models (LMMs). Mass and HLL were strongly related to SVL (rs = 0.866, P < 0.001, and rs = 0.691, P < 0.001, respectively). To avoid collinearity in our model, we used the scaled mass index (Mi) of condition using the following equation: Mi=M×(SVL0/SVL)bSMA

where M and SVL are the body mass and the snout–vent length of the individual, respectively. SVL0 is the mean SVL of the population, and bSMA is the standardized major axis slope from the OLS regression of log-transformed body mass on log-transformed SVL divided by Pearson’s correlation coefficient (Peig & Green, 2009). We also used the relative size of the HLL (hereafter leg) estimated as the residuals of the linear regression between HLL and SVL.

We built linear mixed-effect models (LMMs) using the crossing speed (log-transformed) as response variable, individual as a random factor and Mi, leg, substrate, sex, species and first-order interactions between species and other variables as fixed effects. Because the crossing speed was strongly related to the number of stops in both species (Spearman correlations: T. marmoratus: rs = −0.543, P < 0.001; B. bufo: rs = −0.767, P < 0.001), we also added stops as covariate in our models. We then built a second LMM using stops as response variable, individual as a random factor and Mi, leg, substrate, sex, species and first-order interactions between species and other variables as fixed effects. LMMs were performed using the lme4 R-package (Bates et al., 2017).

Model selection was performed using the Akaike information criteria (AIC; Burnham & Anderson, 2002). If several best models were retained (∆AIC <2), we used a model averaging procedure among all possible models to determine the relative importance of each selected variable. Two parameters from this averaging procedure were retained to test the importance of each variable on the crossing speed: the confidence interval of the averaged estimated slope of the selected term (high effects had confidence intervals that did not include zero) and the relative weight of the term (i.e., the relative Akaike weight of the top-ranked models (ΔAIC <2) in which the term appeared). All statistical analyses were performed using R v.3.1.0 (R Development Core Team, 2014).

Results

After model selection, the best models explaining variation in crossing speed retained substrates (soil and concrete), species, legs, and stops variables (Table 1; Supplementary Material 1). Crossing speed was significantly lower in marbled newts than in common toads, and both species moved faster on soil than on concrete (Table 1; Fig. 1). Crossing speed was also correlated with morphological traits. Long-legged individuals moved faster than individuals with short legs (Table 1).

Table 1 Summary of the averaged linear mixed-effects model showing the influence of significant variables on the crossing speed (in cm/s) for the studied species the marbled newt (Triturus marmoratus; N = 20) and the common toad (Bufo bufo; N = 77).

	Estimate	P	95% CI of estimate	Weight	
(Intercept)	2.2793	<0.001***	(2.2153–2.3433)		
Substrate (soil)	0.0722	<0.001***	(0.0378–0.1067)	1.00	
Species (Triturus marmoratus)	−1.0940	<0.001***	(−1.3533 to −0.8347)	1.00	
leg	0.0998	0.0045**	(0.0309–0.1687)	0.62	
stops	−0.2526	<0.001***	(−0.2754 to −0.2298)	1.00	
Notes:

leg: relative hind-limb length. stops: number of stops. Weight: relative Akaike weight of the top-ranked models (∆AIC < 2) in which the term appeared.

*** P < 0.001,

** P < 0.01.

Figure 1 Mean crossing speed (in cm/s) on concrete and soil substrates in the common toad and the marbled newt.

Crossing speed differed significantly between both substrates in toads (Wilcoxon rank-sum test: P = 0.0101) and in newts (Wilcoxon rank-sum test: P < 0.001). Error bars represent standard error. Notes: *** P < 0.001, ** P < 0.01.

The variable stops was also influenced by species and substrates, and by sex and the interaction between substrates and species (Table 2). In both species, the number of stops was higher in females than in males. In common toads, stops did not vary between substrate types, whereas in marbled newts the number of stops was significantly higher on concrete than on soil (Table 2).

Table 2 Summary of the best linear mixed-effect model showing the influence of significant variables on the number of stops to travel 200 cm during the experimental test for the marbled newt (Triturus marmoratus; N = 20) and the common toad (Bufo bufo; N = 77).

	Estimate	P	
(Intercept)	0.7227	<0.001***	
Substrate (soil)	−0.1212	0.0744	
Species (Triturus marmoratus)	−0.9422	<0.001***	
Sex (male)	−0.5453	0.0009***	
Substrate (soil) * species (Triturus marmoratus)	−0.3937	0.0086**	
Notes:

*** P < 0.001,

** P < 0.01.

Discussion

Our results demonstrated that both species were affected by substrate types, moving significantly slower on a human-made (concrete) than on a natural (soil) substrate. Crossing speed was also related to a morphological trait, with long-legged individuals moving faster than individuals with shorts legs. In addition, we found that the number of stops to cross 200 cm was influenced by sexes, substrates, and species, which indicates that substrates could differently affect the mobility of two sympatric species living into similar habitats.

Influence of substrate type on crossing speed

Inter-patch movement is expected to depend on the nature of the substrate crossed. Some landscape features may be associated with high resistance to movement while others facilitate movement (low resistance). In a previous study, Stevens et al. (2006) experimentally demonstrated that the natterjack toad (B. calamita) significantly preferred substrates mimicking forest and soil than those mimicking agricultural lands. Another similar study revealed that in P. metcalfi, individual crossing speed was higher on asphalt than on grass and soil (Semlitsch et al., 2012) with various consequences on mobility success.

In our experiment, the concrete substrate represented linear roads, both in its nature (mixture of cement and gravel) and length (two m wide road). Roads constitute a very hostile environment for amphibians (dry and warm substrate that could induce a desiccation risk, collisions with vehicles and/or habitat fragmentation; Fahrig et al., 1995; DeMaynadier & Hunter, 2000; Mazerolle, 2004; Marsh et al., 2005; Cosentino, Schooley & Phillips, 2011; Youngquist et al., 2016; Lenhardt et al., 2017). According to our hypotheses, our results showed that substrate type influenced the movement behavior of both species. Contrary to previous studies suggesting that terrestrial amphibians moved more quickly and directly through unfavorable areas in which they are physiologically stressed (Semlitsch et al., 2012; Peterman et al., 2014), we revealed that both common toads and marbled newts moved faster (i.e., higher crossing speed) on soil than on concrete (Fig. 1; Table 1). We then suggest that both species could be more exposed to traffic, and suffer more from both desiccation and mortality risks on roads than on soil (Petronilho & Dias, 2005; Santos et al., 2007; Sillero, 2008; Bouchard et al., 2009; Elzanowski et al., 2009; Matos, Sillero & Argaña, 2012).

In both species, the number of stops was higher in females than in males, independently of substrate type (Table 2). Such differences between sexes may be driven by divergent breeding benefits, which could lead to a trade-off between movement and high energetic costs of reproduction in females, and the well-known sex-biased dispersal (see Trochet et al., 2016b). Indeed, some studies found male-biased dispersal in amphibian species, in anurans as well as in urodela, suggesting that females are philopatric in these organisms (Austin et al., 2003; Lampert et al., 2003; Liebgold, Brodie & Cabe, 2011). In our study, males seem more likely to disperse because they stopped less often than females. As a result, males could also have lower mortality risks induced by collisions with vehicles than females when crossing roads, which may have strong consequences on population dynamics. Our results also demonstrated that the number of stops did not vary between substrates in common toads, whereas marbled newts stopped more often on concrete than on soil (Table 2). This last result shows that marbled newts may be even more at-risk than toads on pavement (and then on roads) because both their crossing speed and their number of stops increased in this substrate type.

In the context of a fragmented landscape, our results corroborate the negative effects of urbanization and human-made infrastructures such as roads on amphibians, leading to an increase in population extinction risk (Fahrig et al., 1995; Carr, Pope & Fahrig, 2002; Sotiropoulos et al., 2013). Here, we highlight a direct influence of the substrate on the displacement of two amphibian species with divergent ecological requirements and locomotion modes living within similar habitats. Our findings are relevant for increased realism of dispersal in predictive modeling, notably by informing landscape permeability for species, body size distribution, and sex-biased dispersal. They also emphasize the importance of road-crossing structure and landscape management at a small spatial scale for amphibian conservation.

Movement-related traits in both species

According to our expectations, our results showed an influence of a morphological trait (leg) on the crossing speed in both species. Various morphological variables enable organisms to be adapted for ecologically effective movement (Bennett, Garland & Else, 1989; Losos, 1990; Choi, Shim & Ricklefs, 2003; Phillips et al., 2006; Moya-Laraño et al., 2008), such as HLL. A recent meta-analysis among several anuran species actually demonstrate that jumping performances were strongly correlated to HLL after correcting by SVL (Gomes et al., 2009). According to this finding, we also demonstrated that movement behavior (defined here as crossing speed) was related to the limb length in both species (see also Bennett, Garland & Else, 1989; Choi, Shim & Ricklefs, 2003; Phillips et al., 2006). Indeed, long-legged individuals moved faster than individuals with short legs (Table 1), which corroborates the idea that limb length may be tightly associated with movement behavior adaptations in amphibians. Longer legs could facilitate more rapid or longer-distance displacement (Phillips et al., 2006), as well as generating other advantages such as improved predator evasion and simplifying the negotiation of barriers and obstacles. As a consequence, the mortality risk of longer-legged individuals could be lower than individuals with short legs.

Limitations of the study

Our experimental protocol is easy-to-use and repeatable but, because of its simplicity, may fail to identify all aspects of the complexity of amphibian’s terrestrial movements. Indeed, our results highlight the impact of substrate nature of human-made infrastructures on amphibian movements at a small spatial scale, but in order to validate and understand the consequences of such an impact, field studies (using capture-mark-recapture or telemetry monitoring) are needed. Also, our studies focus on adults, while most dispersal events are likely to be ensured by juveniles in amphibian species (Semlitsch, 2008). Also, it could be very interesting to test the variation of crossing speed in these organisms under several temperature and humidity conditions. Studying terrestrial displacements of amphibian species on different life stages (larvae, juveniles, and adults) and phases of their life cycle (breeding migrations, dispersal events, etc.) would provide a better identification of the impact of human-made infrastructures on the ecology of amphibian communities, and therefore improve the efficiency of management and conservation efforts.

Conclusions

Inter-patch movement is a multifaceted process, subject to internal and external biotic and abiotic factors. Our findings demonstrate effects of substrates to cross them on the movement behavior in two contrasting amphibian species living within similar habitats. In particular, individuals moved slower on concrete, making them more vulnerable on roads. In both species, we also showed significant relationship between a morphological trait (leg) and crossing speed, suggesting that long-legged amphibians could cross human-made infrastructures faster, which could reduce mortality risk. Comparing the potential influence of various substrates individual movements are essential for explaining and predicting the dynamics of metapopulation living in strongly altered landscapes, which is a prerequisite for developing appropriate conservation management plans.

Supplemental Information

Supplemental Information 1 Crossing speed (in cm/s) depending on substrates (cement or soil) in Triturus marmoratus and Bufo bufo.

SVL: snout-to-vent length; HLL: hind limb length; F: females; M: males; body mass in g; TM: Triturus marmoratus; BB: Bufo bufo; Mi: scaled mass index.

Click here for additional data file.

Supplemental Information 2 Best linear mixed-effects models (in bold) retained by AIC selection showing the significant variables having effect on the crossing speed (in cm/s) for the marbled newt (Tritur us marmoratus; N = 20) and the common toad (Bufo bufo; N = 77).

***: P < 0.001, **: P < 0.01. +: significant positive effect of the factorial variable.

Click here for additional data file.

Additional Information and Declarations

Competing Interests

Author Contributions

Animal Ethics

Field Study Permissions

Data Availability

The authors declare that they have no competing interests.

Audrey Trochet conceived and designed the experiments, analyzed the data, prepared figures and/or tables, authored or reviewed drafts of the paper, approved the final draft.

Hugo Le Chevalier conceived and designed the experiments, performed the experiments, analyzed the data, prepared figures and/or tables, authored or reviewed drafts of the paper, approved the final draft.

Olivier Calvez contributed reagents/materials/analysis tools, approved the final draft.

Alexandre Ribéron conceived and designed the experiments, approved the final draft.

Romain Bertrand analyzed the data, contributed reagents/materials/analysis tools, approved the final draft.

Simon Blanchet conceived and designed the experiments, contributed reagents/materials/analysis tools, approved the final draft.

The following information was supplied relating to ethical approvals (approving body and any reference numbers):

The project was approved by the “Conseil National de la Protection de la Nature” on September 14, 2014 and by the “Conseil Scientifique Régional du Patrimoine Naturel (CSRPN)” of the region Midi-Pyrénées on October 14, 2014. Ethical approval was included under the protected species handling permit from the DREAL Occitanie (“Direction Régionale de l’environnement, de l’Aménagement et du Logement”). Permit nos.: 09-2014-14 and 32-2014-07.

The following information was supplied relating to field study approvals (i.e., approving body and any reference numbers):

Field experiments were approved by the DREAL Occitanie (“Direction Régionale de l’environnement, de l’Aménagement et du Logement”). French permits relating to an authorization of capture, marking, transport, detention, use and release of protected amphibian species have been obtained (permit nos. 09-2014-14 and 32-2014-07).

The following information was supplied regarding data availability:

Raw data is provided in a Supplementary File.

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
