# Peer review of "Influence of substrate types and morphological traits on movement behavior in a toad and newt species"

_PeerJ, doi:10.7717/peerj.6053_

## Round 0.1 · original submission · Major Revisions

We have received two detailed and thoughtful reviews of your manuscript. Both reviewers consider it publishable but raise issues that need to be addressed before publication. Major issues include the choice of statistical analyses and the clarity and reliability of how the statistical results are presented, interpretation and implications of the findings, and the use of English.

I agree with the reviewers and have some additional concerns of my own. You may take my comments below as if they came from a third reviewer: make appropriate changes if you agree with the suggestion and provide a detailed justification if you decide not to make changes. In addition, I have submitted a pdf that highlights errors in English use, grammar and punctuation. There are too many small mistakes for me to explain or correct them all; after revision, please find a native English speaker to go over the entire manuscript before re-submission.

Editor’s Comments

Major Concerns
Objectives and knowledge gap. The goals of the study are not completely clear. On L105-110, you seem to emphasize the interspecific comparison as you did on L31-32 of the Abstract. However, this does not agree with how you presented your findings in the Abstract and how you analyzed the data (species separated with no explicit interspecific comparison) or how you discussed your findings (looking separately at substrate and relative size effects for each species). Furthermore, your study is not designed to strongly investigate questions such as the effect of locomotion mode or body form on crossing movements because you have a sample size of only one species for each form and any differences might be due to many other differences between them. While I accept the importance of fragmentation in general and roads in particular to the survival of amphibian populations and recognize that body form may influence speed of crossing, I do not think you developed the justification for an interspecific comparison of effect of substrate on crossing speed. Finally, the Introduction needs to define the relationship between your study and the previous literature to clearly indicate previous relevant research and the specific knowledge gap addressed by your study. (See also my note below regarding the reference to interspecific interaction.) Please carefully rethink the contributions of your study and coordinate the Abstract, Introduction, Objectives, Statistical Analysis and Discussion to address consistently those contributions.

Statistical analysis. Both reviewers have important questions concerning the statistics, including whether the most appropriate type of analysis was carried out, how the results are presented in the text and tables, and whether the manuscript would benefit by including some additional variables. Reviewer 2 suggests a single overall analysis, including species because of a major objective to compare species. As noted above, I do not place as much weight on this objective because the sample size is so limited for a comparative study. Although I am not particularly sophisticated in biostatistics, I also worry that a single analysis will result in interactions between species and the other variables, raising some interactions from two-way to three-way and potentially obscuring some important and more relevant interspecific patterns.

Presentation of Results. Please double check all your results to be sure that all information in text and tables agrees with your analysis. Reviewer 1 noted that your statement of substrate results was not correct. Reviewer 2 noted there was a problem, possibly a typo, with the description of the effect of stops. On the latter point, the text should indicate the direction of effects. I infer from Table 1 that speed decreased with number of stops in newts and increased with number of stops in toads. If so, it seems that this should be clearly stated and the potential effect of prodding on the two species should be discussed.

Discussion of substrate effects. The Discussion starts with a consideration of substrate effects on movement speed, which I think is appropriate as it is your strongest result. However, there are some major gaps in this section:
• No references are provided for previous studies showing an effect of substrate on locomotion. If it is really true that yours is the first such demonstration, you should highlight that finding. If there have been previous related experiments, you need to cite them.
• Also, give some attention to the effect size. While statistically significant, the effect size appears to be rather small. How does this compare to previous studies, if any, of substrate effects? Is it large enough to be biologically important?
• An important component of a good discussion, is to evaluate the limitations and strengths of your study. For example, might prodding the animals whenever they stopped have had an effect on absolute or relative speed? How might motivating movement by ‘predation’ threat differ from the motivation of moving in nature, which may be migration to the summer home range? How relevant were the substrates used to natural substrates encountered? If amphibians are more likely to cross roads on rainy nights, would testing them on dry 25C days yield relevant measures?
• Finally, consider clarifying the logic of your study in relation to conservation. Roads are already known to be dangerous places for amphibians, with the implication that conservation should minimize exposure to roads. How does the finding that animals move more slowly on roads than on dirt change that? Roads are equally dangerous to crossing amphibians whether or not they move faster on non-road substrates. Would you propose changing the materials used to construct the roads in high-use areas?

Other Concerns
L25, 103. Scientific understanding is not a step in metapopulation function. Rather, it is a step in understanding/explaining/predicting metapopulation function.
L37ff. This statement is closer to a conclusion than a discussion. Discussion should focus on the direct implications and strength of the evidence. I don’t think that the evidence and discussion in the manuscript truly justifies claiming ‘insights into the processes that drive population dynamics’.
L43. Would it be worthwhile from the perspective of literature searches to add ‘connectivity’ and the families of the two study species?
L63. I agree with the reviewer that this manuscript does not address evolutionary ecology and that this sentence is therefore misleading.
L98. Explain what is unclear.
L105-108. It is not clear what you mean by an inter-specific interaction. Are you referring to a statistical interaction? If so, you should be clearer. Many readers will understand an inter-specific interaction as a relationship between two species such as predation or competition.
L149ff. The source of your study animals needs clarification. How many different populations were there? You provide only two geographical locations, but it is not clear if that means that all sites were close to those two locations or something else. You indicate that animals were released in June and July, but do not state when they were captured. Were they captured at breeding sites or away from them? The statement on L164 indicates that they were in the non-breeding terrestrial phase when tested, but it is not clear if you captured them at breeding sites and then allowed them to transition to non-breeding while in captivity. Did you PIT tag and measure the animals immediately before testing or on a previous day? This could potentially influence the response.
L167. For your experiment to be repeatable, you need to provide more information about the test tracks. What precisely was the material? You refer to cement here, but bitumen in the Discussion (L241). This is a huge difference (limestone base vs. petroleum base). Furthermore, cement is usually not used by itself but mixed with gravel to make concrete. The surface texture could be very important, and you should make some attempt to describe how smooth or rough it was. In addition, if freshly made, there could be toxic elements, so readers should know if it was aged. Similarly, what was the composition, surface texture and humidity of the soil? Were the tracks surrounded by a barrier? If so, how high was it? What were the lighting conditions and time of day? What did you poke the animals with? Did you have a set delay before a poke when an animal stopped? Did you clean the tracks between trials?
L170. I presume that crossing speed included stops, but it is important to specify this.
L174-175. This statement is confusing. If each animal had 6 trials on separate days, the minimum for the animals you used would be 6, not 1 day. Did you include some animals without all trials being completed? Furthermore, the reference to another experiment is unclear. Is this intended to explain the longer time period? If so, is it relevant here? I suggest you give the range of number of days over which the 6 trials were accomplished here. Only mention later experiment(s) if it could affect results of this study. If you want to explain captive duration from the perspective of animal welfare, be much clearer about it.
L209. After consistently presenting toads first in Abstract, Introduction, and Methods, the Results addresses newts first. To facilitate reader comprehension, keep the order consistent throughout the manuscript, including figures.
L220-229. This introductory paragraph of the Discussion is excessively redundant of the Introduction. It is appropriate to briefly re-state your findings (L229-232) before you go into detail, but you don’t have to repeat the justification of the study.
L229. It is not clear how substrate preference relates to crossing speed. Do animals spend more time on preferred substrates (i.e. cross more slowly)?
L295-299. I did not see any emphasis on scale effects, nor it is it a direct finding of your study. Furthermore, you did not provide any specific discussion of how comparing one anuran and one urodele provide insights into population dynamics or conservation. This was not clearly raised in the discussion.
Fig. 1. For both caption and figure, note that the SI symbol for seconds is s. Within the caption itself (not as a separate item), provide a sentence identifying the meaning of two asterisks and indicate the test from which the p-value was derived. The numbers on the ordinate do not need decimals. Make sure that the order of species is the same as in the Results text and rest of the manuscript.
Fig. 2. On the ordinate the label should read log crossing speed (cm/s) with the log base included. The caption should also indicate base of the log. Letters for the different panels should match (upper or lower case) the panels.

·

Basic reporting

Introduction – Overall the introduction did a good job of presenting the context for the study – changing landscapes may present new challenges for dispersing/migrating wildlife and the magnitude of impact will depend of specific species requirements. However, I feel that some of the main points were not as clearly described as they could be and with some revision the overall flow and clarity could be improved. Finally, because the experiment compared cement to soil, I think the introduction could benefit from a more narrow focus on the impacts of urbanization and roads on wildlife and movement. There are many studies on movement behavior, physiological responses, and geneflow in response to roads. I also have some specific suggests – see below.

Figures – Figure 1 does not match the description in the text.

Raw Data – Is provided and seems adequate


There are many small English grammar specifics that should be edited. I suggest a thorough reread. For example, from the first paragraph:
Line 52: “Since many decades” should be “For decades”
Line 53: “The anthropogenic practices” should be a more general “Anthropogenic practices”
Line 56: “associated to habitat loss” should be “associated with habitat loss”
Line 57: “ensure a sufficient” should be “ensure sufficient”
Line 58: “could be force” should be “could be forced”
Line 61: “associated to dispersal” should be “associated with dispersal”
Line 64: “pattern in the disturbed landscapes” should be a more general “pattern in disturbed landscapes”

Other specific comments on introduction and results
Abstract:
Line 22: Should be, “Traveling through heterogeneous…”
Line 35-36: I would like more detail about these results.

Introduction:
Line 57-60: This is assuming species already have plasticity or the capacity to adapt their behaviors. It may be that species are already functioning at the maximum of their physiological tolerances and/or dispersal ability. More relevant to your study is that fragmentation and land use change could create inhospitable habitat within the current dispersal range/paths of an individual, rather than them traveling farther.
Line 64: While understanding adaption is a key component of evolutionary biology; the biggest implication from this kind of study is management and conservation. This mention of evolution seems out of place within the context of this paragraph and the paper as a whole.
Line 65: I would rephrase this in the context of complex life histories and be specific for pond- and stream-breeding amphibians. Not all amphibians have aquatic larval stages.
Line 67-68: Terrestrial movements are more risky than what? Aquatic movement? But the risks are the same. Also, do you have a reference for the risk of UV-B exposure during dispersal/migration?
Line 73: One of the reference seems wrong – Youngquist and Boone (2014) looked movement behavior not species richness. Youngquist et al. (2016) looked at distribution and gene flow.
Line 99-108: This part feels more like broad introductory information, not a narrowing of focus to your specific experiment. It is also a little redundant with earlier information. I suggest working the information presented here into earlier paragraphs. I would open this paragraph with your specific objectives.
Line 109-110: When I first read this, I thought you were testing multiple substrates of each category. I recommend specifying that you are testing a single surrogate for each category.
Line 112-114: Do you have references for this information?
Line 115: I need more information about how the “specific ecological requirements” differ between these species.

Results:
Line 210: This figure does not show this.

Experimental design

Research Question – The overall research question is well defined. But I would like to see more specific aims/objectives/hypotheses in the introduction.

Methods – The experimental design is appropriate for the question and the methods are well described.

Specific Comments:
Line 124: Reference?
Line 132: Reference?
Line 149-153: Did you collect individual post-breeding? Or did you have gravid females?
Line 166-167: Were substrates dry or wet? B/c amphibians often move during rain events, how would this affect your results?
Line 167-169: How long did you wait between stops before poking? How did this influence your calculation of crossing speed?
Line 196-203: While not strictly wrong, I would recommend model comparison using AICc. I’m interested to know if you end up with different top models when using AIC vs. backward selection.

Validity of the findings

Results: The data seem robust given the experimental design and the results are valid.

Discussion – The discussion is adequate and generally falls within the scope of the study. I would like to see more discussion on the differences between sexes, within the context of this specific study and sex-biased dispersal. I would also like discussion on the limitations of this kind of laboratory study (e.g., mimicking natural environment and conditions during actual dispersal events, translating lab results to actual in-situ behavior and functional connectivity, using adults vs juveniles, etc). Another consideration is that bare dirt could be considered an anthropogenic land use, i.e. bare ground in agricultural plots. Justification for dirt as a natural ground cover is needed or at least acknowledgement that it is not only found in natural land uses. I would also like greater linkage between the introduction and discussion about roads and road ecology b/c this seems to be the focus on your experiment.

Specific Comments:
Line 223: Why does sensitivity to aquatic changes affect terrestrial dispersal?
Line 276: Were your females gravid or post-breeding? Do you have other references showing differences in dispersal ability between sexes? There are a number of papers on sex-biased dispersal, some amphibian specific (e.g., Austin et al. 2003, Molecular Ecology; Lampert et al. 2003, Molecular Ecology; Palo et al. 2004, Molecular Ecology; Smith and Green 2006, Ecography; Liebgold et al. 2011, Molecular Ecology).
Line 295-296: This is a big idea that seems thrown in. I would like a more extended discussion on the limitations of this kind of study. What evidence is there that laboratory responses correspond to actual responses in the field and functional connectivity?

Additional comments

Overall, this was a well executed study and a well written manuscript. I congratulate the authors on their efforts.

Reviewer 2 ·

Basic reporting

Needs English language editing. Overall concepts can be understood, but choppy to read due to word choice and issues with articles as well as verb-noun agreement. In general, the writing in the Intro and discussion could also be tightened and better organized.

Missing or in appropriate citations in several spots, especially in the introduction. Please see my detailed comments for suggested alternatives or additions.

Article conforms to standard sections and figures are well prepared. The table is visually pleasing, but I would like to see some measure of error (SE), df, and test statistic (F, chi-squared, etc).

Experimental design

How this experiment addresses the knowledge gap could be better defined. Please see my detailed comments for suggestions.

Validity of the findings

I have some concerns about the data analysis.

1. There are lots of problems with this extent of backwards selection. Given the design of your experiment was hypothesis driven, please present the full model. Some might construe this backwards approach as p-hacking.

2. Why were species not considered in the same model? Would only cost 5 df. Seems like that would be the strongest way of getting at your comparative questions. If your short on df’s could consider doing a PCA analysis to compress BI and leg into a single variable.

3. These animals that didn’t cross seem like they would provide valuable information, especially considering they were a sizable proportion of all animals (~25%). If you have multiple measures from these animals, at a minimum test if individuals were more likely to not cross depending on species x substrate (binomial glmm). An even better approach would be to use a glmm with the full data set, and deal with the 0’s by using a poisson / negative binomial / zero-inflated model. With this approach, crossing speed would not be transformed.

4. Likewise, modeling stops could also be interesting and add another layer to your discussion. Also, might want to consider modelling stops as a response variable to look at if stopping rates differ by any of your covariates.

In regard to the conclusions, a major challenge with this sort of study is how to interpret speed. Are animals moving faster because it is a hostile substrate or are they moving faster because the substrate offers lower resistance? This issue, and specifically the potential for ambiguity needs to be addressed at some point.

Additional comments

37: It might be stronger to tie this back to your initial questions regarding movement through heterogeneous landscapes and metapopulation dynamics rather than touting the comparative aspect, which while present, isn’t super strong.

50: “for” instead of “on”

52: “For” instead of “since”

53: “with the development of human activities” is awkward. Rephrase.

53: “These anthropogenic”

55: Habitat

56: associated with

57: While increased connectivity is the result of increased movements, I’d argue individuals aren’t moving more to ensure connectivity. Rather, they may need to move more to find resources or mates
in suboptimal, fragmented landscapes, and a by-product of this is increased connectivity. Please rephrase.

68: This citation does not seem particularly relevant to the previous statement. Please replace. The following may be helpful:

Pittman, S. E., Osbourn, M. S., Drake, D. L., & Semlitsch, R. D. (2013). Predation of juvenile ringed salamanders (Ambystoma annulatum) during initial movement out of ponds. Herpetological Conservation and Biology, 8(3), 681-687.

Pittman, S. E., Osbourn, M. S., & Semlitsch, R. D. (2014). Movement ecology of amphibians: a missing component for understanding population declines. Biological Conservation, 169, 44-53.
Rothermel, B. B., & Luhring, T. M. (2005). Burrow availability and desiccation risk of mole salamanders (Ambystoma talpoideum) in harvested versus unharvested forest stands. Journal of Herpetology, 39(4), 619-626.

73: Regarding gene flow in fragmented landscapes,

Cosentino, B. J., R. L. Schooley, and C. A. Phillips. 2011. Spatial connectivity moderates the effect of predatory fish on salamander metapopulation dynamics. Ecosphere 2(8):art95. doi: 10.1890/ES11-00111.1

Cosentino, B. J., Schooley, R. L., & Phillips, C. A. (2011). Connectivity of agroecosystems: dispersal costs can vary among crops. Landscape ecology, 26(3), 371-379.
77-78: Please add in scientific names for these two species.

92: Also, in amphibians. This study also found no evidence of an effect of leg length in a pond breeding salamander on movement distance.

Ousterhout, B. H., & Semlitsch, R. D. (2018). Effects of conditionally expressed phenotypes and environment on amphibian dispersal in nature. Oikos.

105-107: Seems like Rothermel and Semlitsch 2002 should be cited here. Also, perhaps few studies have simultaneously considered multiple species when looking at movement speed across different substrates, but many studies have used the same methods when testing a single species. Please rephrase to make that clear and cite some of that work here

Rothermel, B. B., & Semlitsch, R. D. (2002). An experimental investigation of landscape resistance of forest versus old‐field habitats to emigrating juvenile amphibians. Conservation biology, 16(5), 1324-1332.

Stevens, V. M., Polus, E., Wesselingh, R. A., Schtickzelle, N., & Baguette, M. (2004). Quantifying functional connectivity: experimental evidence for patch-specific resistance in the Natterjack toad (Bufo calamita). Landscape ecology, 19(8), 829-842.

Semlitsch, R. D., Ecrement, S., Fuller, A., Hammer, K., Howard, J., Krager, C., ... & Walker, M. (2012). Natural and anthropogenic substrates affect movement behavior of the Southern Graycheek Salamander (Plethodon metcalfi). Canadian Journal of Zoology, 90(9), 1128-1135.

114: Please explain briefly here why you predict that differences in movement type (hopping vs walking) might affect responses to different substrates.

127: Migrate, not disperse.

149: Given that juveniles, not adults, are thought to be the primary dispersing stage class, you might want to restructure your intro a bit to introduce the concepts of straying (adults that breed in different locations in different years).

Gamble, L. R., McGarigal, K., & Compton, B. W. (2007). Fidelity and dispersal in the pond-breeding amphibian, Ambystoma opacum: implications for spatio-temporal population dynamics and conservation. Biological Conservation, 139(3-4), 247-257.

Trenham, P. C., Koenig, W. D., & Shaffer, H. B. (2001). Spatially autocorrelated demography and interpond dispersal in the salamander Ambystoma californiense. Ecology, 82(12), 3519-3530.

188: There are some issues with using residual from a linear regression, but the fix does not require much more work. See:

Peig and Green. 2010. The paradigm of body condition: a critical reappraisal of current methods based on mass and length. – Funct. Ecol. 24: 1323–1332.

194: It’s odd the animals that stopped more crossed more quickly. Is this a typo? If not, did poking have a strong effect on speed. If so, that would be a pretty big methodological problem.

208: Which of the six tests listed above does these statistics correspond to? How did you compare crossing speed between soil and cement? Add to methods or results.

227: I’d argue that to truly compare the two species, they need to be considered in the same model.

236: Please reiterate your relative crossing speed finding earlier in this paragraph to remind readers.

275-276: Maybe, but this is not something that would have been detected by your study.

295-296: I agree this is important, but not something addressed in your study.

---

## Round 0.2 · accepted · Accept

Aside from a number of minor grammatical and other corrections, the manuscript is now ready for publication. I have attached a pdf with suggested corrections. One of the statistical tests was missing from the Methods section. After consulting with the corresponding author, I have added a sentence. My sentence is not quite what she suggested. I corrected the grammar and added a reference to the principal test for clarity.

#